# Co-Operative Biofilm Interactions between *Aspergillus fumigatus* and *Pseudomonas aeruginosa* through Secreted Galactosaminogalactan Exopolysaccharide

**DOI:** 10.3390/jof8040336

**Published:** 2022-03-24

**Authors:** Hanna Ostapska, François Le Mauff, Fabrice N. Gravelat, Brendan D. Snarr, Natalie C. Bamford, Jaime C. Van Loon, Geoffrey McKay, Dao Nguyen, P. Lynne Howell, Donald C. Sheppard

**Affiliations:** 1Department of Microbiology and Immunology, McGill University, Montreal, QC H3A 2B4, Canada; hanna.ostapska@mail.mcgill.ca (H.O.); francois.lemauff@mail.mgcill.ca (F.L.M.); fabrice.gravelat@affiliate.mcgill.ca (F.N.G.); brendan.snarr@mail.mcgill.ca (B.D.S.); dao.nguyen@mcgill.ca (D.N.); 2Infectious Disease in Global Health Program, McGill University Health Centre, Montreal, QC H4A 3J1, Canada; 3McGill Interdisciplinary Initiative in Infection and Immunity, Montreal, QC H3A 1Y2, Canada; 4Program in Molecular Medicine, Research Institute, The Hospital for Sick Children, Toronto, ON M5G 0A4, Canada; nbamford001@dundee.ac.uk (N.C.B.); jaime.vanloon@mail.utoronto.ca (J.C.V.L.); howell@sickkids.ca (P.L.H.); 5Department of Biochemistry, University of Toronto, Toronto, ON M5S 1A8, Canada; 6Division of Molecular Microbiology, School of Life Sciences, University of Dundee, Dundee DD1 5EH, UK; 7Meakins-Christie Laboratories, McGill University Health Centre, Montreal, QC H4A 3J1, Canada; geoffrey.mckay@affiliate.mcgill.ca; 8Department of Medicine, McGill University, Montreal, QC H4A 3J1, Canada

**Keywords:** biofilm, *Aspergillus fumigatus*, *Pseudomonas aeruginosa*, exopolysaccharide, Pel, galactosaminogalactan (GAG), interaction, co-operation, resistance, cystic fibrosis

## Abstract

The mold *Aspergillus fumigatus* and bacterium *Pseudomonas aeruginosa* form biofilms in the airways of individuals with cystic fibrosis. Biofilm formation by *A. fumigatus* depends on the self-produced cationic exopolysaccharide galactosaminogalactan (GAG), while *P. aeruginosa* biofilms can contain the cationic exopolysaccharide Pel. GAG and Pel are rendered cationic by deacetylation mediated by either the secreted deacetylase Agd3 (*A. fumigatus*) or the periplasmic deacetylase PelA (*P. aeruginosa*). Given the similarities between these polymers, the potential for biofilm interactions between these organisms were investigated. *P. aeruginosa* were observed to adhere to *A. fumigatus* hyphae in a GAG-dependent manner and to GAG-coated coverslips of *A. fumigatus* biofilms. In biofilm adherence assays, incubation of *P. aeruginosa* with *A. fumigatus* culture supernatants containing de-*N*-acetylated GAG augmented the formation of adherent *P. aeruginosa* biofilms, increasing protection against killing by the antibiotic colistin. Fluorescence microscopy demonstrated incorporation of GAG within *P. aeruginosa* biofilms, suggesting that GAG can serve as an alternate biofilm exopolysaccharide for this bacterium. In contrast, Pel-containing bacterial culture supernatants only augmented the formation of adherent *A. fumigatus* biofilms when antifungal inhibitory molecules were removed. This study demonstrates biofilm interaction via exopolysaccharides as a potential mechanism of co-operation between these organisms in chronic lung disease.

## 1. Introduction

Airway colonization with the mold *Aspergillus fumigatus* and bacterium *Pseudomonas aeruginosa* is common in individuals with chronic lung disease. In individuals with cystic fibrosis, positive respiratory cultures for *A. fumigatus* and *P. aeruginosa* increase in frequency with age [1,2,3]. By adulthood, *P. aeruginosa* is the most common bacterium isolated from respiratory cultures, affecting up to 76% of individuals with cystic fibrosis [2]. *A. fumigatus* is the most common filamentous fungus in this population and is isolated in up to 61% of the adult cystic fibrosis population [2]. The co-isolation of *A. fumigatus* and *P. aeruginosa* has been reported in up to 54% of some cohorts [4]. The co-isolation of these two organisms has been associated with poor clinical outcomes, such that individuals that have persistent *A. fumigatus*–*P. aeruginosa* co-infection exhibit a greater decline in lung function than those infected with either organism alone [4]. The frequent recovery of these two organisms from the airways of patients with chronic lung disease suggests that fungal–bacterial interactions may play a role in the pathogenesis of chronic airway disease.

During a co-culture in vitro, *P. aeruginosa* inhibits the germination of *A. fumigatus* conidia [5]. As a result, the majority of studies examining *P. aeruginosa*–*A. fumigatus* interactions have focused on elucidating the mechanisms underlying this inhibition of fungal growth. Several classes of quorum-sensing molecules, including dirhamnolipids, homoserine lactones, quinolones and phenazines, as well as siderophores secreted by *P. aeruginosa,* can inhibit fungal growth, restrict iron acquisition, or induce oxidative stress. The dirhamnolipid surfactant molecules secreted by *P. aeruginosa* for biofilm maintenance can directly inhibit fungal cell wall *β*-1,3 glucan production, impairing hyphal growth [6,7]. While quinolone-signaling molecules, quinolone 2-heptyl-3-hydroxy-4-quinolone and 2-heptyl-4-quinolone, interfere with conidial germination [8], the mechanism by which homoserine lactone-signaling molecules inhibit fungal growth, and thus biofilm formation is currently unknown [5]. The phenazine 1-hydroxyphenazine, quinolone 2-heptyl-3-hydroxy-4-quinolone, and the extracellular siderophores, pyoverdine and pyochelin, chelate ferric iron, thus restricting fungal access to this important ion and inhibiting growth [9,10,11,12,13,14,15,16]. *P. aeruginosa* can also release an iron-chelating Pf4 bacteriophage that can form an ordered coating on fungal biofilms, thereby limiting hyphal access to iron [17]. At high concentrations, phenazines, including pyocyanin, phenazine-1-carboxylic acid, phenazine-1-carboxamide, as well as phenazine 1-hydroxyphenazine and the siderophore pyochelin, are fungicidal redox-active agents [10,11,12]. Finally, a spectrum of volatile metabolites released by *P. aeruginosa* has been reported to inhibit *Aspergillus* growth, although the mechanisms by which these molecules inhibit fungal grown are not yet well understood [18].

Although these studies are consistent with the fungal growth inhibition by *P. aeruginosa* that is observed in vitro, these findings contrast with the observation that co-colonization with these organisms is commonly reported in patients with chronic lung disease. Therefore, several studies have focused on the mechanisms by which *P. aeruginosa* and *A. fumigatus* could co-operate. While low levels of pyochelin can sequester iron and impair fungal growth, at higher levels this chelator can transfer iron to the fungal siderophore triacetylfusarinine C and stimulate fungal growth [10]. Similarly, sub-inhibitory concentrations of the phenazines pyocyanin, phenazine-1-carboxamide and phenazine-1-carboxylic acid can stimulate fungal growth by facilitating the acquisition of iron by *A. fumigatus* possibly through reducing iron to a more bioavailable ferrous form [11,19,20]. While in in a low iron environment, 2-heptyl-3-hydroxy-4-quinolone is inhibitory to fungal growth, in the presence of high iron levels, this quorum-sensing molecule can also enhance fungal metabolism and growth [14]. Further, volatile metabolic by-products containing sulfur groups released by *P. aeruginosa* have been reported to stimulate fungal growth through interactions with the hyphal cell wall [21,22]. Therefore, the outcome of the interactions between *A. fumigatus* and *P. aeruginosa* may depend on airway microenvironmental factors, including the proximity of the organisms, and local concentrations of diffusible molecules.

Biofilm formation is another process where interactions between *P. aeruginosa* and *A. fumigatus* may occur. The biofilms of *A. fumigatus* and *P. aeruginosa* share a common feature in that both organisms can produce cationic partially de-*N*-acetylated exopolysaccharides that are similar in chemical composition and play important roles in biofilm formation [23,24,25,26,27]. Fungal galactosaminogalactan (GAG) is composed of *α*-1,4-linked D-galactose and partially deacetylated *N*-acetyl-D-galactosamine (GalNAc) and is required for the formation of *A. fumigatus* biofilms on most surfaces [23,24,27]. *P. aeruginosa* Pel polysaccharide mediates biofilm formation by some strains of *P. aeruginosa* [26] and is composed of GalNAc and *N*-acetyl-D-glucosamine, one or both of which are de-*N*-acetylated [28] resulting in a 50% acetylated polymer [29]. Deacetylation plays an important role in the function of both exopolysaccharides by rendering the polymers cationic and adhesive to anionic surfaces [23,25]. The presence of Pel has been demonstrated in *P. aeruginosa* biofilm aggregates in the sputum of patients with cystic fibrosis, suggesting it may play an important role in biofilm formation in vivo [29].

GAG and Pel biosynthesis share several similarities [30]. GAG synthesis begins with the production of the nucleotide monosaccharides uridine diphosphate (UDP)-galactopyranose and UDP-GalNAc by the epimerase Uge3 [23]. These sugars are then polymerized and exported by the putative transmembrane glycosyl transferase Gtb3 [31] where deacetylation of GAG then extracellularly occurs by the secreted deacetylase Agd3 [24]. An Agd3-deficient Δ*agd3* mutant produces fully *N*-acetylated GAG that cannot adhere to hyphae or support biofilm formation [24]. These defects can be restored with the addition of extracellular recombinant Agd3 (rAgd3) [32]. Pel synthesis is similar to GAG in that it is mediated by an inner-membrane multi-protein complex that includes the glycosyl transferase PelF [31,33]. However, Pel is deacetylated within the periplasm through the action of the deacetylase domain of the PelA protein, and is exported as the mature polymer, a process that requires both the membrane-anchored funnel protein PelC and the porin PelB [31,34]. The *P. aeruginosa* PA14 strain is a clinical strain originally isolated from a burn wound [35] that forms Pel-dominant biofilms, as this strain is genetically capable of producing only Pel and not Psl. PA14 is, therefore, the most commonly used isolate for studying Pel-dependent phenotypes [26,36,37]. A *P. aeruginosa* PA14 Δ*pelA* mutant deficient in PelA expression fails to produce secreted Pel; consequently, it lacks the capacity to form biofilms [25]. In contrast, while the clinical isolate and reference strain *P. aeruginosa* PAO1 [38] contains both the Pel and Psl operons, this strain forms Psl-dominant biofilms, and a mutation in *pelA* does not compromise biofilm development [36,39,40]. The similarities in the synthesis and composition of secreted GAG and Pel matrix exopolysaccharides suggests the hypothesis that GAG and/or Pel may mediate co-operative biofilm formation between *A. fumigatus* and *P. aeruginosa*.

Herein, we identified a co-operative interaction between *A. fumigatus* and *P. aeruginosa* at the level of biofilm formation. Confocal and scanning electron microscopy imaging of *P. aeruginosa* with *A. fumigatus* mutant strains during biofilm formation revealed that secreted GAG can mediate *P. aeruginosa* adherence to *A. fumigatus* biofilms. Crystal violet staining demonstrated that wild-type *A. fumigatus* culture supernatants containing secreted GAG can augment the formation of adherent *P. aeruginosa* biofilms. This augmentation is GAG-dependent as culture supernatants from the GAG-deficient Δ*uge3* mutant fungal strain failed to augment adherent bacterial biofilm formation. Culture supernatants containing fully *N*-acetylated GAG from the Δ*agd3* mutant fungal strain augmented *P. aeruginosa* biofilms only when combined with the rAgd3 deacetylase enzyme, suggesting that de-*N*-acetylated GAG is required for this phenotype. Fluorescence confocal microscopy imaging revealed that de-*N*-acetylated GAG incorporates directly into *P. aeruginosa* biofilms, suggesting that GAG may play a structural role in these bacterial biofilms. A functional role for the GAG augmentation of *P. aeruginosa* biofilms was supported by the observation that GAG augmentation protected bacteria against killing by the antibiotic colistin. Although Pel-containing bacterial culture supernatants could augment fungal biofilm biomass adherence, this effect was only detectable when culture supernatants were dialyzed to remove substances that inhibit fungal growth. Collectively, these findings provide a potential mechanism for co-operation between these two organisms within the airways of patients with chronic airway disease.

## 2. Materials and Methods

### 2.1. Fungal and Bacterial Strains and Growth Conditions

*A. fumigatus* and *P. aeruginosa* strains used in this study are detailed in Appendix A. *A. fumigatus* strains were grown on yeast extract-peptone-dextrose (BD Biosciences Difco^TM^, Franklin Lakes, NJ, USA) agar (BD Biosciences Difco^TM^ Bacteriological, Franklin Lakes, NJ, USA) plates at 37 °C, from −80 °C stocks. Conidia were harvested following 6 days of growth with phosphate-buffered saline (PBS, HyClone, Logan, UT, USA) containing 0.1% Tween 80 (Fisher Scientific, Pittsburgh, PA, USA) (PBS-T), washed and resuspended in PBS-T. *A. fumigatus* was grown in Luria-Bertani broth (LB, BD^TM^ Difco^TM^ Miller LB, Franklin Lakes, NJ, USA) or phenol-free RPMI 1640 (Phenol Red-, HEPES-, L-Glutamine-free, Wisent, St-Bruno, QC, Canada) as indicated at 37 °C and in 5% CO_2_. *P. aeruginosa* strains were grown on LB agar plates overnight at 37 °C, from −80 °C stocks. *P. aeruginosa* strains were cultured overnight at 37 °C in LB alone or LB supplemented with 250 μg/mL carbenicillin (Sigma, Burlington, VT, USA) and/or with or without 0.5% L-arabinose (BioShop, Burlington, ON, Canada) as indicated.

### 2.2. Construction of Red Fluorescent Protein (mCherry)-Producing and Green Fluorescent Protein (GFP)-Producing P. aeruginosa Strains

To image interactions with *P. aeruginosa*, bacterial strains were transformed by electroporation as previously described [41], with the pMKB1::*mCherry* plasmid containing the gene encoding the monomeric second-generation red fluorescent protein (RFP) mCherry or by chemical transformation as previously described [42], with the pMKB1::*gfp* plasmid containing the gene encoding the green fluorescent protein (GFP). Transformants were selected by resistance to carbenicillin (250 μg/mL).

### 2.3. Construction of the GFP-Producing A. fumigatus Δuge3 Mutant Strain

To image interactions with the *A. fumigatus* Δ*uge3* mutant strain, *A. fumigatus* Δ*uge3* spheroplasts [43] were transformed with a linearized pGFP-*ble* plasmid encoding GFP and phleomycin resistance. Transformants were screened by resistance to phleomycin (150 μg/mL) [43] and GFP production was confirmed with microscopy.

### 2.4. Galactomannan Quantification in Fungal–Bacterial Interactions by Immunoassay

For conidia co-culture experiments, flasks were inoculated with 1 × 10^6^ *A. fumigatus* conidia/mL in 100 mL LB and 5.5 × 10^7^ *P. aeruginosa* CFU/mL in LB then incubated at 37 °C. For hyphal experiments, fungal cultures were grown for 13 h and then inoculated with bacteria. Culture supernatants were sampled at 18 and 24 h, and galactomannan content of culture supernatants was determined by using the Platelia^TM^ Aspergillus immunoassay kit (Bio-Rad, Hercules, CA, USA), according to the manufacturer’s instructions. Relative fungal growth was determined by comparing the resulting optical density values to an internal standard curve generated using serial dilutions of a pool of lung homogenates from five highly immunosuppressed mice infected with *A. fumigatus* strain Af293, as carried out previously [44,45,46,47,48,49].

### 2.5. Scanning Electron Microscopy 

For studies of direct fungal–bacterial interactions, samples of co-cultured *A. fumigatus* hyphae with *P. aeruginosa* were prepared for scanning electron microscopy using a modification of our previously described dehydration and coating procedure [23]. Hyphae were grown on glass coverslips for 24 h in RPMI 1640. 9.0 × 10^7^ bacterial CFU/mL were co-incubated with hyphae for 0.5 h. Hyphae were washed with PBS and fixed with 2.5% glutaraldehyde (Electron Microscopy Sciences, Hatfield, PA, USA) in 0.1 M calcodylate buffer (BioShop, Burlington, ON, Canada) at room temperature, sequentially dehydrated in ethanol (Commercial alcohols, Toronto, ON, Canada) and then critical-point dried in CO_2_. Coverslips were mounted onto studs and sputter coated with Au-Pd. Hyphae were imaged with field-emission scanning electron microscopy (S-4700 FESEM, Hitachi, Tokyo, Japan) at 10,000× and 5000×.

### 2.6. Confocal Microscopy

For studies of direct fungal–bacterial interactions, cultures of mCherry-producing bacteria supplemented with 250 µg/mL carbenicillin, with or without 0.5% L-arabinose, were co-incubated with GFP-producing hyphae grown in LB as described above; hyphae were washed with PBS and fixed with 1% paraformaldehyde (Electron Microscopy Sciences, Hatfield, PA, USA) for 1 h at 4 °C. Hyphae were washed with PBS and mounted on SlowFade™ (Thermo Fisher Scientific, Waltham, MA, USA), then imaged by confocal microscopy (Zeiss LSM700, Zeiss, Oberkochen, Germany) with 488 and 555 nm lasers at 630×.

For visualization of GAG in bacterial biofilms, co-cultures of *P. aeruginosa* with *A. fumigatus* culture supernatants were grown and stained in a modified version of the previously described chamber coverglass system [50]. An amount of 200 μL of 9.0 × 10^7^ *P. aeruginosa* CFU/mL in LB were grown in 8-well borosilicate glass, non-removable well chambers (Nunc^TM^ Lab-Tec^TM^ II Chambered System, Thermo Fisher, Waltham, MA, USA) with a combination of 640 μL *N*-acetylated-GAG-containing *A. fumigatus* culture supernatants and 165 nM recombinant Agd3 (rAgd3) [32] or LB alone for 24 h. Culture supernatants were then removed and biofilms were stained with a 1:800 dilution of anti-GalNAc antibody (a generous gift from Jean-Paul Latgé, Institute Pasteur, Paris, France) in PBS with 0.2% goat serum (Gibco, Waltham, MA, USA) overnight at room temperature. Biofilms were washed with PBS and further stained with a 1:200 dilution of a secondary Alexa Flour-488 conjugated antibody (Invitrogen^TM^, Waltham, MA, USA) overnight at room temperature, then counter stained with a 1:250 dilution of DRAQ5^TM^ (eBioscience, Waltham, MA, USA) overnight at 4 °C. Biofilms were imaged by confocal microscopy as above. For visualization of GAG with the recombinant carbohydrate-binding module Agd3 (Agd3^141-364^), *P. aeruginosa* biofilms were stained with our previously described GAG binding assay [32]. Biofilms of mCherry-producing *P. aeruginosa* were grown with 250 μg/mL carbenicillin (Sigma, Burlington, VT, USA). Biofilms were stained with 10 μM Alexa Flour-568 labeled Agd3^141-364^ [32] for 2 h at 4 °C, washed twice with PBS and fixed with 4% paraformaldehyde. Biofilms were imaged by confocal microscopy as above. Mean fluorescent intensity was determined with ImageJ (version 1.52a) software (National Institutes of Health, Bethesda, MD, USA).

### 2.7. Culture Supernatant Collection

Fungal culture supernatants were prepared in flasks inoculated with 1 x 10^6^ *A. fumigatus* conidia/mL in 100 mL LB and incubated at 37 °C [23]. After 26 h, culture supernatants were filtered through either a 0.22 μm nylon membrane (Steritop^TM^, Waltham, MA, USA) or Miracloth (Millipore Sigma, Burlington, ON, Canada). Culture supernatants were either lyophilized (Labconco, Kansas City, MO, USA) or stored at −20 °C.

Bacterial culture supernatants were prepared in flat-bottom, non-tissue, culture-treated, 24-well plates (Falcon, Burlington, VT, USA) inoculated with 2 mL of 9.0 × 10^7^ *P. aeruginosa* CFU/mL in LB per well with or without 0.5% L-arabinose (BioShop, Burlington, ON, Canada) and statically incubated at 37 °C. 48 h old culture supernatants were aspirated, separated from bacteria by centrifugation (Sorvall Legend RT+, Thermo Scientific, Waltham, MA, USA) at 2643× *g* for 30 min, and residual planktonic bacteria were heat-killed for 1 h at 60 °C [6]. Culture supernatants were dialyzed through 3.5 kDa molecular weight cut-off cellulose membrane (Fisherbrand, Pittsburgh, PA, USA or Spectrum Labs, Waltham, MA, USA) in 2 passes of deionized water, 1 pass of deionized water with 0.2% sodium azide (BioShop, Burlington, ON, Canada), 3 additional passes of deionized water, and 3 passes of 0.1% PBS. Culture supernatants were then either lyophilized or stored at 4 °C.

### 2.8. Crystal Violet Biomass Adherence Assay

For studies of bacterial biofilm adherence, adherent biofilm biomass was quantified using a modification of the previously described assay [51]. Bacterial biofilms were grown in flat-bottom, non-tissue, culture-treated, 96-well plates (Falcon, Burlington, VT, USA) that were inoculated with 100 μL of 9.0 × 10^7^ *P. aeruginosa* CFU/mL in LB and diluted 1:2 with a combination of 50 μL LB and 50 μL of *A. fumigatus* or *P. aeruginosa* culture supernatants or LB alone. For biofilm deacetylation studies, 130 nM rAgd3 was added to plates that were prepared as described above. Plates were incubated for 22 h at 37 °C and then washed with 300 μL deionized water and stained with 200 μL 0.1% crystal violet (BioShop, Burlington, ON, Canada) for 10 min. Crystal violet was solubilized with 200 μL ethanol (Commercial alcohols, Toronto, ON, Canada) for 10 min, and the absorbance of 150 μL of the solution was measured at 600 nm (Tecan Infinite M200PRO, Tecan, Mannedorf, Switzerland).

For studies of fungal biofilm adherence, adherent biofilm biomass was quantified using a modification of our previously described crystal violet assay [42]. Fungal biofilms were grown in round-bottom, non-tissue, culture-treated, 96-well plates (Falcon, Burlington, VT, USA) that were inoculated with 1 × 10^6^ *A. fumigatus* conidia/mL in 50 μL LB and 50 μL of *P. aeruginosa* or *A. fumigatus* culture supernatants or LB alone and incubated for 20 h at 37 °C in 5% CO_2_. Plates were washed with 100 μL deionized water and stained with 100 μL 0.1% crystal violet for 10 min. Crystal violet was solubilized with 100 μL ethanol for 10 min, and the absorbance of the resulting solution was measured as above.

### 2.9. Antibiotic Susceptibility Assay

For antimicrobial challenge of augmented adhered bacterial biomass, bacterial biofilms were grown in the presence of fungal culture supernatants as described above. Antimicrobial treatment was performed using a modified version of our previously described antibiotic susceptibility assay [51]. Briefly, 22 h old biofilms were washed with PBS and treated with 250 μL of 1.17 μg/mL colistin sulfate salt (Sigma, Burlington, VT, USA) in PBS for 5 h at 37 °C in 5% CO_2_. Bacterial viability was measured using a modification of a previously described XTT tetrazolium salt (2,6-bis-(2-methoxy-4-nitro-5-sulfophenyl)-2H-tetrazolium-5-caboxanilide) metabolic assay [52]. Briefly, 50 μL XTT tetrazolium salt solution (100 μg/mL XTT tetrazolium salt (BioShop, Burlington, ON, Canada) and 25 μM menadione (Sigma, Burlington, VT, USA) was added to wells and plates were incubated at 37 °C in the dark for 2 h. The absorbance of 180 μL of the solution was measured at 450 nm (Tecan Infinite M200PRO, Tecan, Mannedorf, Switzerland).

### 2.10. GAG Quantification by Enzyme-Linked Lectin Assay (ELLA)

GAG was quantified in fungal culture supernatants using a modified version of our previously described ELLA [32]. *N*-acetylated or de-*N*-acetylated GAG-containing *A. fumigatus* culture supernatants were serially diluted in TRIS-buffered saline (TBS, Alfa Aesar, Haverhill, MA, USA) in 96-well Immulon 4HBX plates (Thermo Scientific, Waltham, MA, USA). *N*-acetylated GAG-containing culture supernatants were de-*N*-acetylated with 130 nM rAgd3 for 1 h at room temperature. Plates were washed 3 times with TBS with 0.05% Tween 20 (T20, BioShop, Burlington, ON, Canada), stained with lectin solution (30 nM biotinylated soybean agglutinin (Vector Labs, Burlingame, CA, USA) and a 1:700 dilution of avidin conjugated to horseradish peroxidase (Invitrogen)) at room temperature for 1 h. Plates were washed 3 times with TBS-T20 and developed with 3,3′,5,5′-tetramethybenzidine (TMB) substrate ultrasensitive solution (Millipore, Burlington, VT, USA) and 1 M H_2_SO_4_ (BioShop, Burlington, ON, Canada) stop solution. Absorbance was measured at 450 nm (Tecan Infinite M200PRO, Tecan, Mannedorf, Switzerland).

### 2.11. Statistical Analysis

Data are presented and statistical significance is calculated as indicated. All graphs were generated, and statistical analyses were performed in GraphPad Prism (version 9.0.0) software (GraphPad, San Diego, CA, USA). Significant differences between values were compared by one-way analysis of variance (ANOVA) with Tukey’s multiple-comparison test, paired *t* test and unpaired *t* test as noted in the figure legends.

## 3. Results

### 3.1. A. fumigatus Hyphae Are More Resistant Than Conidia to the Inhibitory Effects of P. aeruginosa

To determine the effect of *P. aeruginosa* on the growth of different *A. fumigatus* morphologies, bacteria were co-cultured with either *A. fumigatus* conidia or hyphae, and relative fungal growth was monitored by quantifying the relative levels of a secreted fungal exopolysaccharide, galactomannan, over time by immunoassay. Galactomannan is constitutively produced by *A. fumigatus* hyphae and has been used as a surrogate to quantify the growth kinetics of hyphae, which correlate to fungal burden in vivo [42,44,45,46,47,48,49,53,54,55]. Even after 24 h of growth, only low levels of galactomannan were detected in co-cultures of *P. aeruginosa* with *A. fumigatus* conidia, consistent with prior reports that *P. aeruginosa* strongly inhibits *A. fumigatus* conidial germination (Figure 1a) [5]. In contrast, when *P. aeruginosa* were added to pre-grown *A. fumigatus* hyphae (13 h), no significant decrease in galactomannan production was observed between hyphae cultured with and without bacteria at up to 24 h after bacterial inoculation (Figure 1b). These results suggest that *A. fumigatus* hyphae are more resistant than conidia to the inhibitory effects of *P. aeruginosa.*

### 3.2. GAG but Not Pel Is Required for Adherence of P. aeruginosa to A. fumigatus Biofilms

The ability of *A. fumigatus* hyphae to persist in the presence of *P. aeruginosa* suggests that bacteria could interact with fungal biofilms produced by *A. fumigatus* hyphae within co-infected lungs. To begin to probe the direct interactions between fungal biofilms and bacteria, confocal microscopy was used to image biofilms produced by green fluorescent protein (GFP)-producing wild-type *A. fumigatus* co-cultured with the red-fluorescent protein (mCherry)-producing wild-type *P. aeruginosa* strain PA14. Confocal imaging of these co-cultures after washing revealed the presence of abundant *P. aeruginosa* adherent to fungal biofilms, with notable clustering of bacterial cells surrounding *A. fumigatus* hyphae (Figure 2a). Pel-deficient Δ*pelA P. aeruginosa* exhibited similar levels of adherence to wild-type *A. fumigatus* biofilms as the wild-type bacteria (Figure 2a,b). However, both wild-type *P. aeruginosa* and Pel-deficient Δ*pelA P. aeruginosa* were found to poorly adhere to GAG-deficient Δ*uge3 A. fumigatus* hyphae (Figure 2c,d). Taken together, these findings suggest that GAG, and not Pel, is required for the adhesion of *P. aeruginosa* to fungal hyphae within biofilms. To test if the inability of Pel to compensate for the loss of GAG was a consequence of insufficient levels of Pel production by *P. aeruginosa* PA14, the ability of a *P. aeruginosa* PAO1 Pel-overexpressing Δ*wspF*Δ*psl*P_BAD_*pel* mutant strain (which also lacks the ability to produce the Psl polysaccharide) to adhere to GAG-deficient hyphae was tested. As with wild-type bacteria, the Pel-overexpressing Δ*wspF*Δ*psl*P_BAD_*pel* strain adhered well to wild-type fungal biofilms (Figure 2e). However, Pel-overproduction did not augment bacterial adherence to GAG-deficient Δ*uge3* hyphae (Figure 2f). Collectively, these findings suggest that GAG, but not Pel, mediates *P. aeruginosa* adherence to *A. fumigatus* biofilms. 

To confirm these findings and better define the localization of *P. aeruginosa* within fungal biofilms, the scanning electron microscopy of fungal biofilms co-cultured with *P. aeruginosa* was performed. The scanning electron microscopy examination of the wild-type *P. aeruginosa* PA14 strain co-cultured with wild-type *A. fumigatus* biofilms confirmed the presence of bacteria directly adherent to fungal hyphae within fungal biofilms (Figure 3a). As was observed by confocal microscopy, the adherence of *P. aeruginosa* to *A. fumigatus* hyphae appeared to be dependent on the production of fungal GAG and not Pel, as wild-type *P. aeruginosa* and Pel-deficient Δ*pelA P. aeruginosa* were rarely observed bound to GAG-deficient Δ*uge3* hyphae (Figure 3b,c), whereas wild-type and Pel-deficient *P. aeruginosa* were both observed to bind in large numbers to wild-type *A. fumigatus* hyphae (Figure 3a,d). These data are consistent with the hypothesis that fungal GAG is necessary for the adherence of *P. aeruginosa* to hyphae within fungal biofilms. 

In addition to the population of hyphal-bound bacteria, wild-type and Pel-deficient Δ*pelA P. aeruginosa* organisms were also observed to bind to coverslips in the interstitial space between wild-type *A. fumigatus* hyphae (Figure 3a,d). In contrast, few wild-type *P. aeruginosa* were observed on the coverslips of the biofilm-deficient Δ*uge3* hyphae (Figure 3e). These findings suggest that, in addition to binding to *A. fumigatus* hyphae, *P. aeruginosa* may adhere to secreted biofilm components that are adherent to the surface of coverslips.

### 3.3. Bacterial-Adherent Biofilm Formation Is Augmented by Secreted Fungal Products

Given the observation that the adherence of *P. aeruginosa* to an abiotic surface was enhanced with products secreted into the biofilm matrix by *A. fumigatus* hyphae, we hypothesized that products secreted by *A. fumigatus* could augment the formation of adherent *P. aeruginosa* biofilms. To test this hypothesis and quantify this phenomenon, culture supernatants were collected from *A. fumigatus* and co-cultured with *P. aeruginosa* in static culture in round bottom non-tissue culture-treated 96-well plates (biofilm-forming conditions). Adherent biofilm formation was quantified with crystal violet staining. *P. aeruginosa* co-cultured with *A. fumigatus* culture supernatants exhibited increased adherent biofilm formation compared with bacteria cultured in media alone (Figure 4a). Given that *P. aeruginosa* adhered poorly to coverslips in the presence of the GAG-deficient *A. fumigatus* Δ*uge3* mutant (Figure 3e), we hypothesized that secreted GAG was the biofilm matrix product that was responsible for the fungal-mediated augmentation of adherent *P. aeruginosa* biofilm formation. Consistent with this hypothesis, culture supernatants from the GAG-deficient Δ*uge3* mutant failed to increase *P. aeruginosa*-adherent biofilm formation (Figure 4a). The presence of GAG in wild-type culture supernatants was confirmed by a GAG-enzyme-linked lectin assay (Appendix A). Culture supernatants from wild-type *A. fumigatus*, but not the GAG-deficient Δ*uge3* mutant, also resulted in increased the adherent biofilm formation of the Pel-deficient Δ*pelA P. aeruginosa* mutant, suggesting that Pel polysaccharide is not required for the GAG-mediated augmentation of adherent bacterial biofilm formation (Figure 4b). 

The inability of culture supernatants from the GAG-deficient *A. fumigatus* Δ*uge3* mutant to augment *P. aeruginosa* adherent biofilm formation suggests that secreted GAG is responsible for the observed fungal-mediated effects on *P. aeruginosa* biofilm. This observation also suggests that the fungal-mediated augmentation of wild-type *P. aeruginosa* adherence is not a consequence of increased deacetylation of Pel polysaccharide by the secreted fungal deacetylase Agd3, as the Δ*uge3* mutant secretes active Agd3 (112). As purified GAG is insoluble, to confirm that Agd3 alone cannot directly increase adherent bacterial biofilm formation, wild-type *P. aeruginosa* and the Pel deacetylase-deficient Δ*pelA* mutant were grown in the presence of recombinant Agd3 (rAgd3). As predicted, in comparison to bovine serum albumin protein, rAgd3 treatment did not increase adherent biofilm formation by either bacterial strain (Appendix A). Collectively, these data suggest that the fungal-mediated augmentation of *P. aeruginosa* adherent biofilm formation is mediated via GAG polysaccharide rather than the activity of the fungal deacetylase on Pel polysaccharide.

To determine if the deacetylation of GAG is required for augmentation of bacterial adherent biofilm formation, the Pel-deficient Δ*pelA* mutant was co-cultured with culture supernatants from the Agd3-deficient Δ*agd3 A. fumigatus* mutant, which produces only fully *N*-acetylated GAG [24]. In contrast to de-*N*-acetylated GAG-containing wild-type *A. fumigatus* culture supernatants, *A. fumigatus* culture supernatants containing fully *N*-acetylated-GAG failed to augment *P. aeruginosa*-adherent biofilm formation (Figure 5). Together, these results suggest that de-*N*-acetylated, rather than *N*-acetylated GAG, is the secreted product in wild-type *A. fumigatus* culture supernatants that augments adherence of the bacterial biofilms.

### 3.4. Secreted GAG Integrates into the Architecture of P. aeruginosa Biofilms

To determine if de-N-acetylated GAG augments the adherence of bacterial biofilm biomass through direct incorporation into the biofilm matrix, adherent wild-type *P. aeruginosa* biofilms treated with the combination of rAgd3 and N-acetylated-GAG-containing culture supernatants were stained with a monoclonal anti-GAG antibody [27]. Immunofluorescence microscopy demonstrated that this antibody stained both wild-type and GAG-treated *P. aeruginosa*-adherent biofilms (Figure 6a), suggesting that this antibody likely detects α-1,4-linked GalNAc regions that are shared by both polymers. However, GAG-treated *P. aeruginosa*-adherent biofilms exhibited significantly higher staining than wild-type *P. aeruginosa* biofilms (mean fluorescence intensity of 39.3 vs. 18.1, respectively), suggesting that de-N-acetylated GAG may incorporate into the *P. aeruginosa* biofilm architecture (Figure 6a).

To confirm that the observed increase in fluorescence intensity detected with anti-GalNAc staining was a consequence of incorporating GAG into the biofilm matrix, rather than from the upregulation of Pel production, we exploited the specificity of Agd3 binding to GalN/GalNAc regions of the GAG polymer [32]. Native and GAG-treated *P. aeruginosa* biofilms were incubated with a fluorophore-labeled recombinant carbohydrate binding module of Agd3 (Agd3^141^^-^^364^) and then imaged with confocal microscopy to detect GAG. Consistent with our GalNAc antibody staining results, fluorescence microscopy revealed Agd3^141^^-^^364^ binding to GAG-treated, GFP-producing, wild-type *P. aeruginosa* biofilms (mean fluorescence intensity of 48.0) (Figure 6b). In contrast, wild-type *P. aeruginosa* biofilms and those grown in the presence of fully *N*-acetylated-GAG-containing culture supernatants did not bind Agd3^141^^-^^364^, confirming the GAG specificity of Agd3^141^^-^^364^ (mean fluorescence intensity of 4.8 and 10.5, respectively) (Figure 6b). Collectively, these results suggest that the fungal augmentation of bacterial-adherent biofilms is a result of the incorporation of the GAG polymer as a structural element into the biofilm architecture, rather than of GAG-mediated induction of Pel production. 

### 3.5. GAG-Mediates Resistance to Colistin within P. aeruginosa Biofilms

Pel in Pel-dependent *P. aeruginosa* biofilms has been demonstrated to enhance resistance to killing by subinhibitory concentrations of the antibiotic colistin [51]. To determine if GAG incorporation into the biofilm matrix can also enhance the resistance of bacterial biofilms to antimicrobials, the sensitivity of wild-type and GAG-treated *P. aeruginosa* biofilms to colistin was compared. Wild-type and GAG-treated *P. aeruginosa*-adherent biofilms were treated with colistin, and bacterial survival was measured by the quantification of XTT tetrazolium salt metabolism. Consistent with the results of crystal violet experiments (Figure 4a), which suggest that GAG treatment increases the biomass of adherent bacterial biofilms, an increase in bacterial metabolic activity of GAG-treated, adherent *P. aeruginosa* biofilms was observed in comparison to that of wild-type bacterial biofilms (Figure 7). GAG-treated, adherent *P. aeruginosa* biofilms biomass exhibited higher levels of resistance to colistin than wild-type biofilms (decrease of 28% vs. 37% in metabolic activity, respectively) (Figure 7). These data suggest that the secreted GAG that was incorporated into *P. aeruginosa* biofilms could protect against antimicrobial killing at least as well as, if not better than, native bacterial exopolysaccharide.

### 3.6. P. aeruginosa Secretes Growth-Inhibitory Products That Counteract the Augmentation of A. fumigatus-Adherent Biofilm Formation by Secreted Pel

Given that secreted de-*N*-acetylated GAG can augment the formation of adherent *P. aeruginosa* biofilms, we sought to determine if the opposite was true and if secreted Pel polysaccharide could augment the formation of adherent fungal biofilms. As with GAG, purified Pel is insoluble, and to test this hypothesis, young hyphae (8 h) of wild-type and GAG-deficient Δ*uge3* mutant *A. fumigatus* were co-cultured with culture supernatants from Pel-overproducing Δ*wspF*Δ*psl*P_BAD_*pel P. aeruginosa*. Pel-containing *P. aeruginosa* culture supernatants failed to increase the formation of adherent biofilms by either fungal strain (Figure 8a). Indeed, co-culture with *P. aeruginosa* culture supernatants inhibited wild-type *A. fumigatus* biomass accumulation, consistent with reports of the inhibition of *A. fumigatus* growth and subsequent biofilm formation by a range of small molecules secreted by *P. aeruginosa*. These findings may reflect an inability of Pel to augment adherent *A. fumigatus* biofilm formation, or that the inhibitory effects of small molecules present in *P. aeruginosa* culture supernatants [5,6,7,8,9,10,11,17] dominate over any effects that Pel may have on augmenting fungal biofilm formation. Purified Pel is not available; therefore, we were unable to directly test if purified Pel could augment adherent *A. fumigatus* biofilm formation in the absence of bacterial inhibition of fungal growth. *P. aeruginosa* culture supernatants were, therefore, dialyzed to remove inhibitory soluble metabolites prior to co-culture with *A. fumigatus*. The treatment of the GAG-deficient Δ*uge3 A. fumigatus* mutant with dialyzed culture supernatants from Pel-overproducing *P. aeruginosa* resulted in increased adherent fungal biofilm formation, albeit to a lesser extent than treatment with GAG-containing culture supernatants (Figure 8b). The augmentation of adherent fungal biofilm formation was not observed when *A. fumigatus* was co-cultured with dialyzed *P. aeruginosa* PA14 culture supernatants, suggesting that augmentation of fungal biofilm formation by Pel is dose dependent (Figure 8b). Taken together, these results suggest that while Pel can compensate for the loss of GAG exopolysaccharide and augment adherent biofilm formation by the GAG-deficient mutant, under standard in vitro conditions, the biofilm inhibitory effects of *P. aeruginosa*-secreted small molecules dominate.

## 4. Discussion

Herein, we describe the potential for co-operative biofilm interactions between *A. fumigatus* and *P. aeruginosa*. The contributions of both fungi and bacteria to this interaction were investigated with imaging and biofilm adherence analyses. These studies support previous reports that *A. fumigatus* hyphal GAG mediates the direct binding between these organisms [6] and further elucidates that this interaction is Pel-independent. The dissection of indirect interactions between these species identified the potential for co-operative biofilm interactions mediated by secreted exopolysaccharides. The secreted *A. fumigatus* exopolysaccharide, GAG, was found to physically integrate into bacterial biofilms and could augment resistance to the antibiotic colistin. In contrast, the effects of secreted *P. aeruginosa* products were dominated by the anti-fungal activity of *P. aeruginosa*-secreted products that counteract the Pel-mediated enhancement of *A. fumigatus* biofilms [5,6,8,10,11,16,56,57,58]. Therefore, this work suggests a potential unidirectional co-operativity in biofilm formation between these organisms.

This study found that hyphal-associated GAG was required for the direct adherence of *P. aeruginosa* to *A. fumigatus*, while the production of bacterial Pel was dispensable for these interactions. A previous study identified the requirement of GAG for the direct interaction of the *P. aeruginosa* PAO1 strain with *A. fumigatus* [59]. However, the *P. aeruginosa* PAO1 strain used in that study predominantly secretes a neutrally charged exopolysaccharide Psl and produces only low levels of Pel [28,36,39,60,61]. Our studies, using Pel-producing, Psl-deficient PA14 and Pel-overexpressing strains, demonstrated that even the overproduction of Pel was insufficient to support the adherence of bacterial cells to *A. fumigatus* hyphae in the absence of GAG. Therefore, direct contact between these organisms is uniquely dependent on GAG.

The resistance of biofilm-forming organisms to antimicrobials is, in part, a consequence of the formation of an exopolysaccharide-containing matrix that impedes the penetration of these molecules [62]. Biofilm exopolysaccharides with a cationic charge can contribute to biofilm resistance to antimicrobials through charge–charge interactions within the matrix [62]. GAG within *A. fumigatus* biofilms mediates resistance to killing by antifungals, likely by repelling cationic molecules or hindering the cellular uptake of large nonpolar molecules [42]. The Pel exopolysaccharide of *P. aeruginosa*, which has a similar chemical structure to GAG, mediates the resistance of *P. aeruginosa* biofilms to the cationic antibiotic colistin, possibly by charge–charge interactions [51]. GAG was found to incorporate into *P. aeruginosa* biofilms, and GAG-augmented bacterial biofilms exhibited an increased resistance to colistin. This observation suggests that the incorporation of a fungal exopolysaccharide did not destabilize the architecture and function of *P. aeruginosa* biofilms. Given the similarities in structure between Pel and GAG, it is plausible that cationic GAG may also mediate resistance to cationic colistin through charge–charge mediated repulsion. 

Secreted products from *A. fumigatus* and *P. aeruginosa* have the potential to have pleiotropic effects on the other organism in co-cultures. As of now, studies have largely focused on mutually antagonistic interactions in vitro [5,6,8,9,11,13,17,63]. *A. fumigatus* strains, including the wild-type strain (Af293) used in this study, secrete the mycotoxin gliotoxin that is toxic to *P. aeruginosa* and can be detectable in fungal culture supernatants at as early as 24 h of growth in vitro [64,65]. However, *A. fumigatus* GAG-containing culture supernatants were found to augment the formation of adherent biofilms by the *P. aeruginosa* PA14 strain, suggesting that the ability of GAG to augment bacterial biofilm formation is dominant over any inhibitory effects due to gliotoxin or other toxic secondary metabolites. Similarly, *P. aeruginosa* secrete a plethora of small molecules that have been observed to inhibit *A. fumigatus* growth and biofilm formation [5,6,8,10,11,56]. Consistent with these findings, we found that Pel-containing *P. aeruginosa* cultures’ supernatants failed to augment *A. fumigatus*-adherent biofilm formation. The ability of dialyzed culture supernatants from the Pel-overexpressing strain to complement adherent biofilm formation by the GAG-deficient *A. fumigatus* Δ*uge3* mutant suggests that, while Pel can act as a surrogate exopolysaccharide for a mutant strain deficient in biofilm adherence under specific conditions, this effect is easily masked by the dominant inhibitory effects of other bacterial secreted factors. Similar evidence for the production of competing compounds by *P. aeruginosa,* which enhance and inhibit fungal growth, was found when the co-operative fungal growth-enhancing effects of bacterial-derived volatile molecules were revealed by physical barrier co-cultures that prevented soluble anti-fungal molecules from contacting *A. fumigatus* in vitro [21,22]. 

The predominance of studies identifying antagonistic interactions between *P. aeruginosa* with *A. fumigatus* contrasts with the clinical observation that these organisms commonly co-colonize the airways of individuals with chronic lung disease [5,6,9,10,11,17,56,66]. Our findings provide a potential mechanism whereby these two species that exhibit antagonistic interactions in vitro can conditionally co-operate. A similar co-operative inter-species interaction between *P. aeruginosa* and *C. albicans* has been described at the level of bacterial exopolysaccharide expression. While *P. aeruginosa* directly antagonizes *C. albicans* growth by killing hyphae and indirectly by suppressing filamentation, *C. albicans* is often co-isolated with *P. aeruginosa* in up to 29% of individuals with cystic fibrosis [67,68,69]. A metabolite-driven positive feedback loop between bacterial phenazine and *Candida* ethanol production, that was shown to upregulate Pel expression and, consequently, enhance biofilm formation by *P. aeruginosa*, has been postulated to underlie the co-operativity between these two species [70]. Collectively, these studies suggest the possibility that the fungal–bacterial modulation of biofilm adherence or formation may play an important role in the persistence of pulmonary microorganisms in chronic lung disease.

The complex nature of the potential for inhibitory and enhancing interactions between *A. fumigatus* and *P. aeruginosa* suggests that the outcome of the interactions between *A. fumigatus* and *P. aeruginosa* may depend on the microenvironment in which these organisms reside. Further work in animal models of chronic airway co-infection will, therefore, be required to determine the potential significance of the interactions that have been documented in vitro. Currently, however, no model of chronic, non-invasive *Aspergillus*/*Pseudomonas* co-colonization exists to support these studies. Although chronic airway co-colonization with both organisms has been reported using organisms embedded in agar or other matrix beads [67,69], this model is not suitable to study indirect biofilm interactions, as the use of an exogenous matrix is a potentially major confounder of exopolysaccharide interactions. The findings of our study highlight the need for ex vivo or chronic co-infection models to better dissect *A. fumigatus* and *P. aeruginosa* interactions during chronic airway co-infection.

Our studies demonstrate that a secreted fungal exopolysaccharide can augment adherent bacterial biofilm formation and resistance to antibiotic killing in vitro. Co-operativity may contribute to the worsening of pathogenesis and outcomes in co-colonized individuals. This mechanism of co-operative interactions should be explored in vivo with the development of a relevant and robust co-infection model that can inform treatment outcomes and lead to novel therapeutic strategies.

## Figures and Tables

**Figure 1 jof-08-00336-f001:**
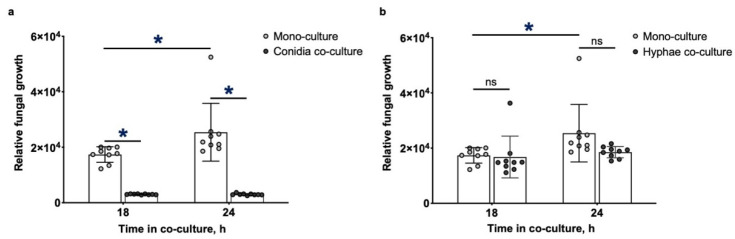
*A. fumigatus* hyphae are more resistant than conidia to the growth inhibitory effects of *P. aeruginosa* during co-culture. (**a**) Growth of *A. fumigatus* conidia in co-culture with *P. aeruginosa* and (**b**) growth of *A. fumigatus* pre-grown hyphae in co-culture with *P. aeruginosa* was monitored by quantifying relative galactomannan (GM) over time using a GM enzyme immunoassay (EIA). Bars represent the means ± standard deviations of 3 independent experiments. A significant increase in growth is indicated by * (*p* < 0.05) and no significant difference in growth is indicated by ns as determined by unpaired *t* test.

**Figure 2 jof-08-00336-f002:**
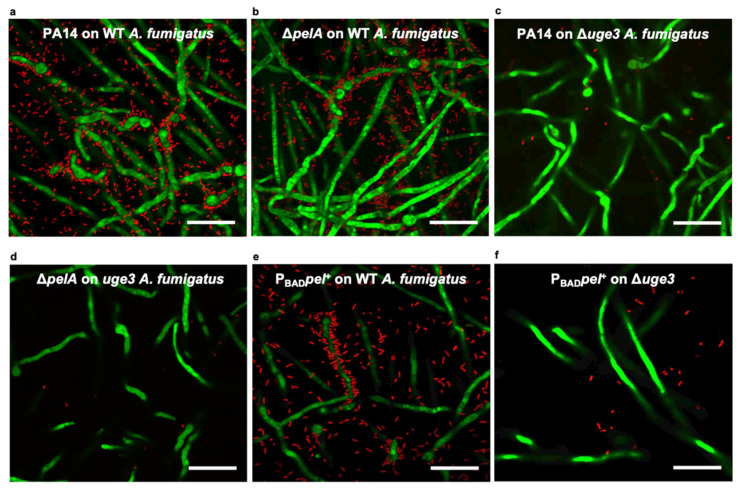
GAG but not Pel is required for the adherence of *P. aeruginosa* to *A. fumigatus*. Confocal microscopy images of green fluorescent protein (GFP)-producing *A. fumigatus* co-cultured with mCherry-producing *P. aeruginosa*. (**a**) Wild-type *P. aeruginosa* (PA14) adherent to hyphae of wild-type (WT) *A. fumigatus*. (**b**) Pel-deficient *P. aeruginosa* (Δ*pelA*) adherent to hyphae of WT *A. fumigatus*. (**c**) Wild-type *P. aeruginosa* (PA14) not adherent to hyphae of GAG-deficient *A. fumigatus* (Δ*uge3*). (**d**) Pel-deficient *P. aeruginosa* (Δ*pelA*) not adherent to hyphae of GAG-deficient *A. fumigatus* (Δ*uge3*). (**e**) Pel-overexpressing Δ*wspF*Δ*psl*P_BAD_*pel P. aeruginosa* (P_BAD_*pel*^+^) adherent to hyphae of WT *A. fumigatus*. (**f**) Pel-overexpressing Δ*wspF*Δ*psl*P_BAD_*pel P. aeruginosa* (P_BAD_*pel*^+^) not adherent to hyphae of GAG-deficient *A. fumigatus*. Representative images of at least 2 independent experiments. Imaged at 630× (scale bar, 20 μm).

**Figure 3 jof-08-00336-f003:**
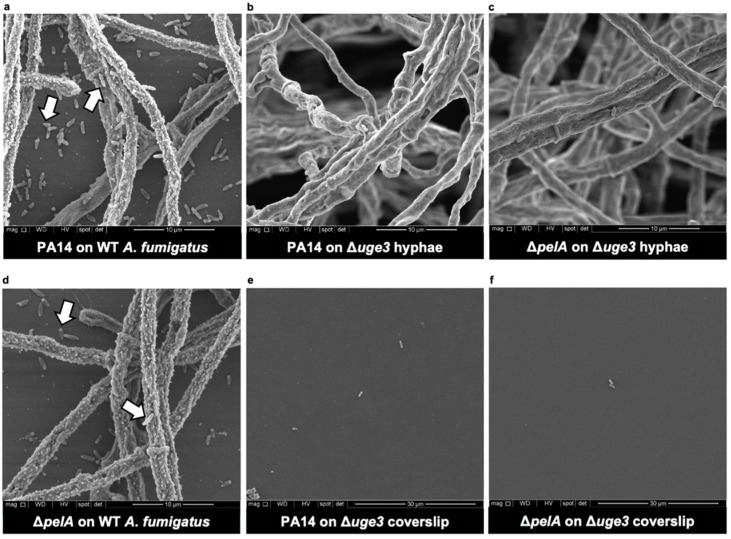
GAG mediates the adherence of *P. aeruginosa* to *A. fumigatus* hyphae. Scanning electron microscopy images of co-cultures of wild-type *P. aeruginosa* (PA14) with (**a**) adherent wild-type (WT) *A. fumigatus* hyphae on coverslips or (**b**) non-adherent GAG-deficient *A. fumigatus* hyphae (Δ*uge3*). Co-cultures of Pel-deficient *P. aeruginosa* (Δ*pelA*) with (**c**) non-adherent GAG-deficient *A. fumigatus* hyphae (Δ*uge3*) or with (**d**) adherent wild-type *A. fumigatus* hyphae (WT) on coverslips. (**e**) Coverslip of (**b**) and (**f**) coverslip of (**c**). Representative images of 2 independent experiments. (**a**–**d**) Imaged at 10,000×. (**e**,**f**) Note a lower magnification at 5000×. White arrows indicate adherent bacteria to hyphae and coverslips in the interstitial spaces between hyphae.

**Figure 4 jof-08-00336-f004:**
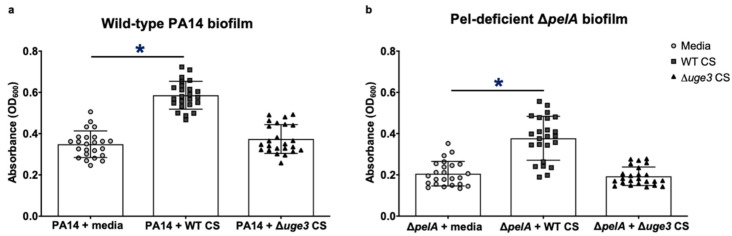
*A. fumigatus*-secreted products increase the formation of *P. aeruginosa*-adherent biofilms. Crystal violet quantification of adherent biofilm biomass by (**a**) wild-type *P. aeruginosa* (PA14) and (**b**) Pel-deficient *P. aeruginosa* (Δ*pelA*) grown in the presence of media control, wild-type *A. fumigatus* culture supernatants (WT CS), or GAG-deficient *A. fumigatus* culture supernatants (Δ*uge3* CS). Bars represent the means ± standard deviations of the destain solution measured at 600 nm of 3 independent experiments. A significant increase in absorbance is indicated by * (*p* < 0.0001) relative to all bars as determined by one-way ANOVA with Tukey’s multiple-comparison test.

**Figure 5 jof-08-00336-f005:**
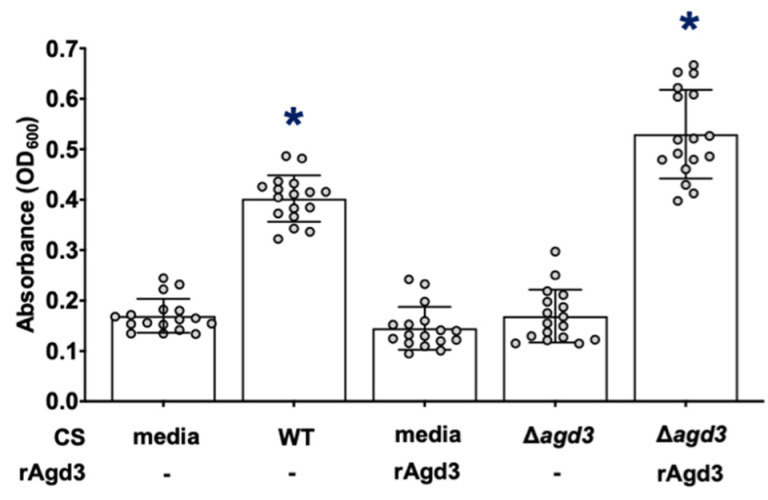
De-*N*-acetylated GAG is required to increase adherent biofilm formation by Pel-deficient *P. aeruginosa*. Adherent biofilm formation by Pel-deficient *P. aeruginosa* (Δ*pelA*) grown in the presence of de-*N*-acetylated GAG from WT de-*N*-acetylated GAG-containing CS (WT CS), 130 nM recombinant Agd3 deacetylase enzyme (rAgd3) or *N*-acetylated GAG from *N*-acetylated-GAG-containing culture supernatants (Δ*agd3* CS), or de-*N*-acetylated GAG from a combination of *N*-acetylated-GAG-containing culture supernatants and 130 nM rAgd3 (Δ*agd3* CS + rAgd3) was quantified with crystal violet staining. Bars represent the means ± standard deviations of 3 independent experiments. A significant increase in absorbance is indicated by * (*p* < 0.0001) relative to all bars as determined by one-way ANOVA with Tukey’s multiple-comparison test.

**Figure 6 jof-08-00336-f006:**
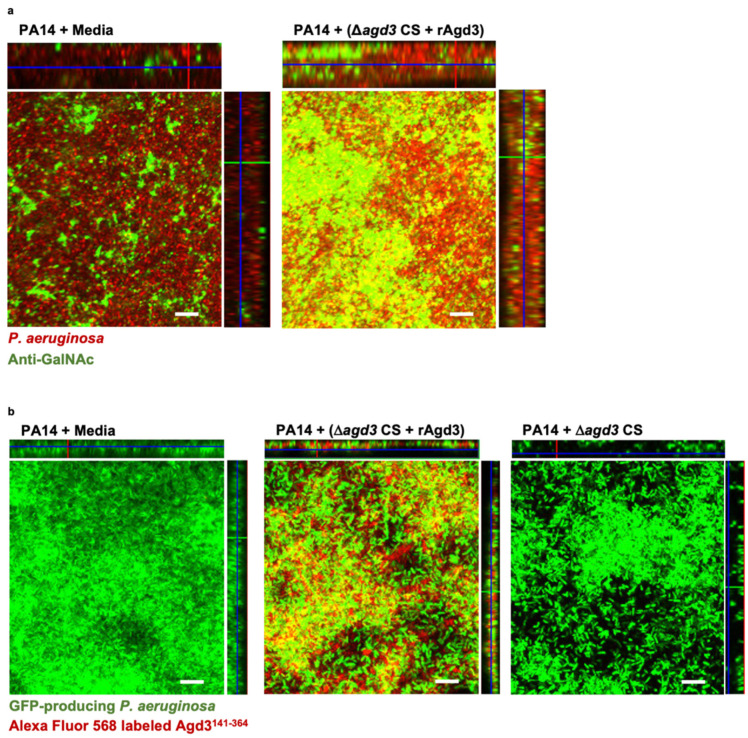
GAG incorporates into *P. aeruginosa* biofilms. (**a**) Confocal immunofluorescence microscopy maximal intensity projections with orthogonal projections of wild-type *P. aeruginosa* (PA14) biofilms grown in the presence of de-*N*-acetylated GAG from a combination of *N*-acetylated GAG-containing culture supernatants and 165 nM recombinant Agd3 deacetylase (Δ*agd3* CS + rAgd3) or media. Exopolysaccharide was detected by staining with anti-*N*-acetyl-D-galactosamine antibody (anti-GalNAc) and an Alexa Flour-488 conjugated anti-rabbit secondary antibody (green). *P. aeruginosa* was counter stained with DRAQ5 (red). (**b**) Confocal microscopy maximal intensity projections with orthogonal projections of wild-type green fluorescent protein (GFP)-producing (green) *P. aeruginosa* (PA14) biofilms grown in the presence of de-*N*-acetylated GAG from a combination of Δ*agd3* CS and 165 nM rAgd3, Δ*agd3* CS alone or in media. Biofilms were imaged with fluorescence microscopy. Exopolysaccharide was detected by staining with Alexa Flour-568-labeled recombinant carbohydrate binding module Agd3^141-364^ (red). Representative images of 3 independent experiments. Biofilms were imaged at 630× (scale bar, 10 μm).

**Figure 7 jof-08-00336-f007:**
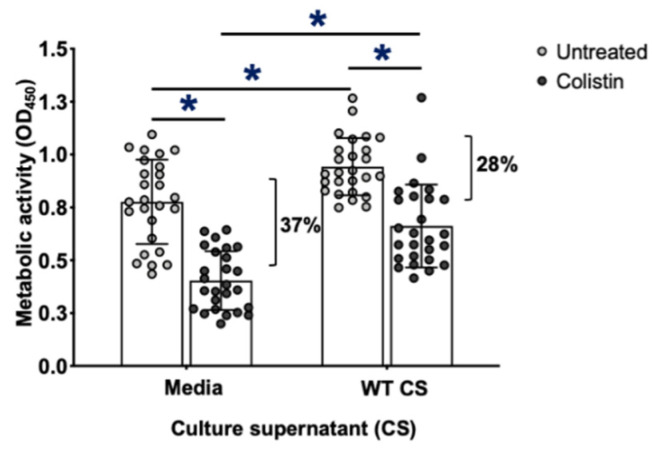
Adherent GAG-treated *P. aeruginosa* biofilms exhibit increased resistance to colistin. Biofilms of wild-type *P. aeruginosa* PA14 were growth in the presence of de-*N*-acetylated GAG-containing culture supernatants from wild-type *A. fumigatus* culture supernatants (WT CS) and then treated with 0.00117 mg/mL of colistin. Viability of *P. aeruginosa* was determined by XTT tetrazolium salt metabolism. Bars represent the means ± standard deviations of XTT supernatant measured at 450 nm of 5 independent experiments. A significant decrease in absorbance is indicated by * (*p* < 0.0005) between untreated and treated groups and between treated groups and as determined by paired *t* test.

**Figure 8 jof-08-00336-f008:**
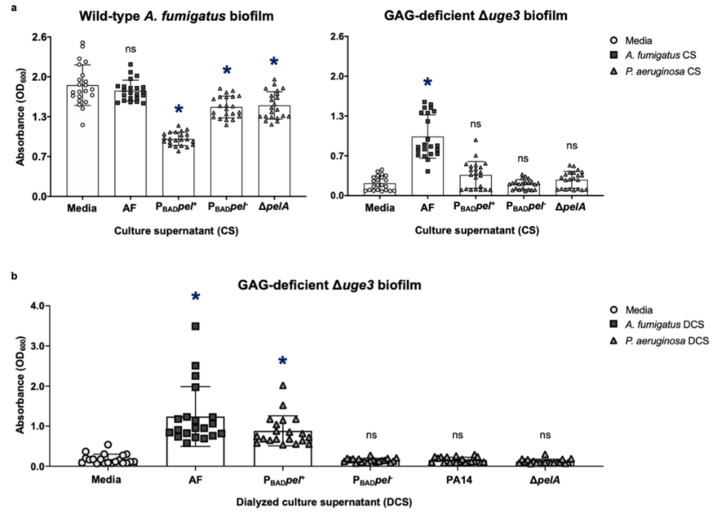
Inhibitory effects on the growth of *A. fumigatus* by secreted products within *P. aeruginosa* culture supernatants that dominate over the augmentation of *A. fumigatus*-adherent biofilm formation by secreted Pel. Adherent biofilm formation by (**a**) wild-type *A. fumigatus* (AF) (left) and GAG-deficient *A. fumigatus* hyphae (Δ*uge3*) (right) grown in the presence of culture supernatants (CS) from wild-type *A. fumigatus* (AF), Pel-overexpressing Δ*wspF*Δ*psl*P_BAD_*pel P. aeruginosa* (P_BAD_*pel*^+^), Pel-nonexpressing Δ*wspF*Δ*psl*P_BAD_*pel P. aeruginosa* (P_BAD_*pel*^-^), or Pel-deficient *P. aeruginosa* (Δ*pelA*). (**b**) GAG-deficient *A. fumigatus* hyphae (Δ*uge3*) grown in the presence of dialyzed culture supernatants (DCS) from wild-type *A. fumigatus* (AF), Pel-overexpressing Δ*wspF*Δ*psl*P_BAD_*pel P. aeruginosa* (P_BAD_*pel*^+^), Pel-nonexpressing Δ*wspF*Δ*psl*P_BAD_*pel P. aeruginosa* (P_BAD_*pel*^−^), wild-type Pel-dominant *P. aeruginosa* (PA14), or Pel-deficient *P. aeruginosa* (Δ*pelA*) was quantified with crystal violet staining. Bars represent the means ± standard deviations of the destain solution measured at 600 nm of at least 3 independent experiments. A significant decrease or increase in absorbance is indicated by * (*p* < 0.01), and no significant difference in absorbance is indicated by ns relative to cultures grown in media alone as determined by one-way ANOVA with Dunnett’s multiple-comparison test.

## Data Availability

Not applicable.

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
