# Peer review of "Co-Operative Biofilm Interactions between Aspergillus fumigatus and Pseudomonas aeruginosa through Secreted Galactosaminogalactan Exopolysaccharide"

_jof, 2022, doi:10.3390/jof8040336_

Round 1
Reviewer 1 Report
The manuscript #jof-1624713, entitled “Co-operative biofilm interactions between Aspergillus fumigatus and Pseudomonas aeruginosa through secreted galactosaminogalactan exopolysaccharide” by Ostapska et al. presents a comprehensive study on, in line with the title, the mutual influence of P. aeruginosa and A. fumigatus in biofilm development. The manuscript is very well presented - I appreciate logical design of the study, logical design of the manuscript itself, as well as aesthetics such as good quality figures and almost none editorial mistakes. Also, I appreciate the fact that all necessary informations are included in the introduction and discussion sections. Both sections have anticipated all of my questions, which originated during exploring the manuscript.
Author Response
The manuscript #jof-1624713, entitled “Co-operative biofilm interactions between Aspergillus fumigatus and Pseudomonas aeruginosa through secreted galactosaminogalactan exopolysaccharide” by Ostapska et al. presents a comprehensive study on, in line with the title, the mutual influence of P. aeruginosa and A. fumigatus in biofilm development. The manuscript is very well presented - I appreciate logical design of the study, logical design of the manuscript itself, as well as aesthetics such as good quality figures and almost none editorial mistakes. Also, I appreciate the fact that all necessary informations are included in the introduction and discussion sections. Both sections have anticipated all of my questions, which originated during exploring the manuscript.
Thank you for your kind comments!
Reviewer 2 Report
The manuscript by Ostapska et al. describes a study where a component of P. aeruginosa bacteria, pelB, is shown to have a minor effect on the coexistence of bacteria and a fungal pathogen A. fumigatus. On the contrary, the GAG component of the fungus is shown to have a critical role in the coexistence of the two species in cell culture models. Since the literature data were provided where the second component of P. aeruginosa polysaccharide, psL, is present together with the pelB in bacteria, the work of Ostapska with psL-deficient strain and subsequent genetic manipulation of P. aeruginosa showed clearly that the literature data has to be corrected.
The work is done very well and the presentation is very clear. The antibody staining slides, however, may be convincing visually but the data would be enhanced if it could be supported by quantifying the area positive vs negative for antibody staining. Also, the quality of pictures for the antibody staining is lower than the rest of the data. It could be caused by the aperture (?) or system setting rendering such quality.
Author Response
The manuscript by Ostapska et al. describes a study where a component of P. aeruginosa bacteria, pelB, is shown to have a minor effect on the coexistence of bacteria and a fungal pathogen A. fumigatus. On the contrary, the GAG component of the fungus is shown to have a critical role in the coexistence of the two species in cell culture models. Since the literature data were provided where the second component of P. aeruginosa polysaccharide, psL, is present together with the pelB in bacteria, the work of Ostapska with psL-deficient strain and subsequent genetic manipulation of P. aeruginosa showed clearly that the literature data has to be corrected.
The work is done very well and the presentation is very clear. The antibody staining slides, however, may be convincing visually but the data would be enhanced if it could be supported by quantifying the area positive vs negative for antibody staining. Also, the quality of pictures for the antibody staining is lower than the rest of the data. It could be caused by the aperture (?) or system setting rendering such quality.
We added the mean fluorescent intensity for panel b in the text as we had with panel a and specified the software used to quantify pixel intensity in the methods, lines 452, 454-455 and 247-248. The apparent difference in resolution may reflect the different methods used to visualize the bacteria in panels a (propidium iodide which stains intracellular nucleic acid) and b (plasmid GFP expression by bacteria), and exopolysaccharide in panels a (antibody) and b (labeled protein).
Reviewer 3 Report
It was a pleasure to read this paper and I want to complement the authors with a very nice piece of research showing how A. fumigatus and P. aeruginosa interact in biofilms and the role of secreted exopolysaccharides. This specific co-infection is a huge burden to patients with CF, and to manage these co-infections properly, more insight is needed about the molecular mechanisms underpinning the pathophysiology. The work presented is solid with nice pictures of microscopy.
I only have a few comments for the authors to be addressed:
Line 104: this reads as only a minority of the P. aeruginosa strains do form biofilms by employing Pel; this is important as if this is true, how does that relate to the generalisation of their results; is there any data that the P. aeruginosa strains infecting CF patients are more or less likely to be employing Pel?
Line 122: the PA14 strain used in this study, is this a clinical strain derived from a CF patient? Is the PA14 strain a good representative strain of the P. aeruginosa strains found in CF patients? The authors may want to add a bit more information on this in the intro and/or discussion.
Paragraph 2.4: it is not clear how the hyphal experiments were performed, the authors state that fungal cultures were inoculated with bacteria after 13hr of growth, do they mean that the flasks as mentioned in line 183 were incubated for 13 hrs before the bacteria were added? If so, that would not be defined as a biofilm, but as hyphal growth in liquid media.
Is it possible to provide an estimation of the number of bacteria instead of only providing the OD?
Line 312: how long where the hyphae ‘pre-grown’?
Figure 1: the fungal growth (arb. Unit) needs to be explained, how does this relates to GM levels?
Line 363: ‘biofilm-forming condition’ > it is not directly clear from the methodology what a biofilm-forming condition is, and what is considered a mature biofilm; can the authors make this a bit more clear for the reader?
Line 451: what are ‘young hyphae’, for how long were they grown?
Author Response
It was a pleasure to read this paper and I want to complement the authors with a very nice piece of research showing how A. fumigatus and P. aeruginosa interact in biofilms and the role of secreted exopolysaccharides. This specific co-infection is a huge burden to patients with CF, and to manage these co-infections properly, more insight is needed about the molecular mechanisms underpinning the pathophysiology. The work presented is solid with nice pictures of microscopy.
I only have a few comments for the authors to be addressed:
Line 104: this reads as only a minority of the P. aeruginosa strains do form biofilms by employing Pel; this is important as if this is true, how does that relate to the generalisation of their results; is there any data that the P. aeruginosa strains infecting CF patients are more or less likely to be employing Pel?
This is a very good question. Pel production by P. aeruginosa has been demonstrated in clinical sputum samples from CF patients. Although a comprehensive survey of the frequency of Pel production has not been conducted, in this study it was found in 5 out of 5 patients, now lines 109-111.
Line 122: the PA14 strain used in this study, is this a clinical strain derived from a CF patient? Is the PA14 strain a good representative strain of the P. aeruginosa strains found in CF patients? The authors may want to add a bit more information on this in the intro and/or discussion.
Thank you for the question. PA14 is a clinical isolate that was originally isolated from a burn wound however it is the most common laboratory strain used for studying Pel, now lines 124-126. In addition to the study cited above, we have also recently characterized several CF respiratory isolates that produce Pel at the same or higher level than PA14 (unpublished data).
Paragraph 2.4: it is not clear how the hyphal experiments were performed, the authors state that fungal cultures were inoculated with bacteria after 13hr of growth, do they mean that the flasks as mentioned in line 183 were incubated for 13 hrs before the bacteria were added? If so, that would not be defined as a biofilm, but as hyphal growth in liquid media.
Thank you for the comment, we agree that this section was ambiguous. Yes, the fungus was incubated in flasks for 13 hours which were then inoculated with bacteria, now line 192. While we agree these are not classic biofilms, we and others have shown that under shaking conditions, fungal hyphae form into aggregates (fungal balls). Hyphae within these balls are covered in GAG as are hyphae in sessile biofilms.
Is it possible to provide an estimation of the number of bacteria instead of only providing the OD?
Thank you for the comment. We converted the optical densities to the number of colony forming units per millimeter, lines 191, 205, 231-232, 258, 276.
Line 312: how long where the hyphae ‘pre-grown’?
Thank you for the question. The fungi were pre-grown for 13 hours, now line 335.
Figure 1: the fungal growth (arb. Unit) needs to be explained, how does this relates to GM levels?
Thank you for the question, we agree that this section needed clarification. We have shown that GM determination by EIA can serve as a surrogate to measure fungal growth kinetics in vivo. However, the correlation curve of GM index to fungal burden is not linear and a standard curve is required. Unfortunately, GM is not commercially available for this purpose. Therefore, as we have done in our previous work, relative GM levels were determined by comparison with a standard curve generated by serial dilutions of a pool of lung homogenates from five immunosuppressed mice infected with A. fumigatus strain Af293, now lines 189, 195-199, 331-334 and Figure 1 legend.
Line 363: ‘biofilm-forming condition’ > it is not directly clear from the methodology what a biofilm-forming condition is, and what is considered a mature biofilm; can the authors make this a bit more clear for the reader?
Thank you for the comment. In the biofilm-forming condition, fungi were grown in static culture in round bottom non-tissue culture-treated 96-well plates, now lines 391-392.
Line 451: what are ‘young hyphae’, for how long were they grown?
Good eye! In our assay young hyphae were grown for 8 hours, now line 482.